# What do Neural Machine Translation Models Learn about Morphology?

## Abstract

Neural machine translation (MT) models obtain state-of-the-art performance while maintaining a simple, end-to-end architecture. However, little is known about what these models learn about source and target languages during the training process. In this work, we analyze the representations learned by neural MT models at various levels of granularity and empirically evaluate the quality of the representations for learning morphology through extrinsic part-of-speech and morphological tagging tasks. We conduct a thorough investigation along several parameters: word-based vs. character-based representations, depth of the encoding layer, the identity of the target language, and encoder vs. decoder representations. Our data-driven, quantitative evaluation sheds light on important aspects in the neural MT system and its ability to capture word structure.

## 1 Introduction

Neural network models are quickly becoming the predominant approach to machine translation (MT). Training neural MT (NMT) models can be done in an end-to-end fashion, which is simpler and more elegant than traditional MT systems. Moreover, NMT systems have become competitive with, or better than, the previous state-of-the-art, especially since the introduction of sequence-to-sequence models and the attention mechanism (Bahdanau et al., 2014; Sutskever et al., 2014). The improved translation quality is often attributed to better handling of non-local dependencies and morphology generation (Luong and Manning, 2015; Bentivogli et al., 2016).

However, little is known about what and how much these models learn about each language and its features. Recent work has started exploring the role of the NMT encoder in learning source syntax (Shi et al., 2016), but research studies are yet to answer important questions such as: *(i)* what do NMT models learn about word morphology? *(ii)* what is the effect on learning when translating into/from morphologically-rich languages? *(iii)* what impact do different representations (character vs. word) have on learning? and *(iv)* what do different modules learn about the syntactic and semantic structure of a language? Answering such questions is imperative for fully understanding the NMT architecture. In this paper, we strive towards exploring *(i)*, *(ii)*, and *(iii)* by providing quantitative, data-driven answers to the following specific questions:

- Which parts of the NMT architecture capture word structure?

- What is the division of labor between different components (e.g. different layers or encoder vs. decoder)?

- How do different word representations help learn better morphology and modeling of infrequent words?

- How does the target language affect the learning of word structure?

To achieve this, we follow a simple but effective procedure with three steps: *(i)* train a neural MT system on a parallel corpus; *(ii)* use the trained model to extract feature representations for words in a language of interest; and *(iii)* train a classifier using extracted features to make predictions for another task. We then evaluate the quality of the trained classifier on the given task as a proxy to the quality of the extracted representations. In

this way, we obtain a quantitative measure of how well the original MT system learns features that are relevant to the given task.

We focus on the tasks of part-of-speech (POS) and full morphological tagging. We investigate how different neural MT systems capture POS and morphology through a series of experiments along several parameters. For instance, we contrast word-based and character-based representations, use different encoding layers, vary source and target languages, and compare extracting features from the encoder vs. the decoder.

We experiment with several languages with varying degrees of morphological richness: French, German, Czech, Arabic, and Hebrew. Our analysis reveals interesting insights such as:

- Character-based representations are much better for learning morphology, especially for low-frequency words. This improvement is correlated with better BLEU scores. On the other hand, word-based models are sufficient for learning the structure of common words.

- Lower layers of the MT encoder are better at capturing word structure, while higher layers are more focused on word meaning.

- The target language impacts the kind of information learned by the MT system. Translating into morphologically-poorer languages leads to better source-side word representations. This is partly, but not completely, correlated with BLEU scores.

- The neural decoder learns very little about word structure. The attention mechanism removes much of the burden of learning word representations from the decoder.

## 2 Methodology

Given a source sentence $s = \{w_1, w_2, ..., w_N\}$ and a target sentence $t = \{u_1, u_2, ..., u_M\}$, we first generate a vector representation for the source sentence using an encoder (Eqn. 1) and then map this vector to the target sentence using a decoder (Eqn. 2) (Sutskever et al., 2014):

$$\text{ENC} : s = \{w_1, w_2, ..., w_N\} \mapsto \mathbf{s} \in \mathbb{R}^k \quad (1)$$

$$\text{DEC} : \mathbf{s} \in \mathbb{R}^k \mapsto t = \{u_1, u_2, ..., u_M\} \quad (2)$$

In this work, we use long short-term memory (LSTM) (Hochreiter and Schmidhuber, 1997)

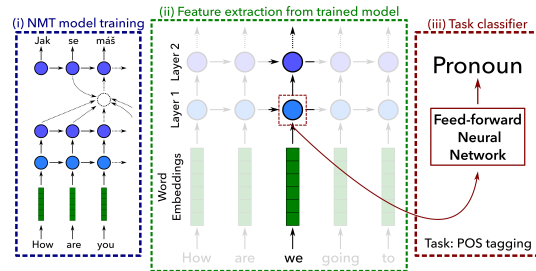

Figure 1: Illustration of our approach: (i) NMT system trained on parallel data; (ii) features extracted from pre-trained model; (iii) classifier trained using the extracted features. Here a POS tagging classifier is trained on features from the first hidden layer.

encoder-decoders with attention (Bahdanau et al., 2014), which we train on parallel data.

After training the NMT system, we freeze the parameters of the encoder and use ENC as a feature extractor to generate vectors representing words in the sentence. Let $\text{ENC}_i(s)$ denote the encoded representation of word $w_i$. For example, this may be the output of the LSTM after word $w_i$. We feed $\text{ENC}_i(s)$ to a neural classifier that is trained to predict POS or morphological tags and evaluate the quality of the representation based on our ability to train a good classifier. By comparing the performance of classifiers trained with features from different instantiations of ENC, we can evaluate what MT encoders learn about word structure. Figure 1 illustrates this process. We follow a similar procedure for analyzing representation learning in DEC.

The classifier itself can be modeled in different ways. For example, it may be an LSTM over outputs of the encoder. However, as we are interested in assessing the quality of the representations learned by the MT system, we choose to model the classifier as a simple feed-forward network with one hidden layer and a ReLU non-linearity. Arguably, if the learned representations are good, then a non-linear classifier should be able to extract useful information from them.[1] We emphasize that our goal is not to beat the state-of-the-art on a given task, but rather to analyze what NMT models learn about morphology. The classifier is trained with a cross-entropy loss; more details on its architecture are in the supplementary material.

---

[1] We also experimented with a linear classifier and observed similar trends to the non-linear case, but overall lower results; Qian et al. (2016b) reported similar findings.

|  | Ar | De | Fr | Cz |
|---|---|---|---|---|
|  | Gold/Pred | Gold/Pred | Pred | Pred |
| Train | 500K/2.7M | 888K/4.0M | 5.2M | 2.0M |
| Dev | 63K/114K | 45K/50K | 55K | 35K |
| Test | 62K/16K | 44K/25K | 23K | 20K |
| POS Tags | 42 | 54 | 33 | 368 |
| Morph Tags | 1969 | 214 | – | – |

Table 1: Statistics for annotated corpora in Arabic (Ar), German (De), French (Fr), and Czech (Cz).

## 3 Data

**Language pairs** We experiment with several language pairs, including morphologically-rich languages, that have received relatively significant attention in the MT community. These include Arabic-, German-, French-, and Czech-English pairs. To broaden our analysis and study the effect of having morphologically-rich languages on both source and target sides, we also include Arabic-Hebrew, two languages with rich and similar morphological systems, and Arabic-German, two languages with rich but different morphologies.

**MT data** Our translation models are trained on the WIT[3] corpus of TED talks (Cettolo et al., 2012; Cettolo, 2016) made available for IWSLT 2016. This allows for comparable and cross-linguistic analysis. Statistics about each language pair are given in Table 1 (under Pred). We use official dev and test sets for tuning and testing. Reported figures are the averages over test sets.

**Annotated data** We use two kinds of datasets to train POS and morphological classifiers: gold-standard and predicted tags. For predicted tags, we simply used freely available taggers to annotate the MT data. For gold tags, we use gold-annotated datasets. Table 1 gives statistics for datasets with gold and predicted tags; see supplementary material for details on taggers and gold data. We train and test our classifiers on predicted annotations, and similarly on gold annotations, when we have them. We report both results wherever available.

## 4 Encoder Analysis

Recall that after training the NMT system we freeze its parameters and use it only to generate features for the POS/morphology classifier. Given a trained encoder ENC and a sentence $s$ with POS/morphology annotation, we generate word features $\text{ENC}_i(s)$ for every word in the sentence.

|  | Gold | Pred | BLEU |
|---|---|---|---|
|  | Word/Char | Word/Char | Word/Char |
| Ar-En | 80.31/93.66 | 89.62/95.35 | 24.7/28.4 |
| Ar-He | 78.20/92.48 | 88.33/94.66 | 9.9/10.7 |
| De-En | 87.68/94.57 | 93.54/94.63 | 29.6/30.4 |
| Fr-En | – | 94.61/95.55 | 37.8/38.8 |
| Cz-En | – | 75.71/79.10 | 23.2/25.4 |

Table 2: POS accuracy on gold and predicted tags using word-based and character-based representations, as well as corresponding BLEU scores.

We then train a classifier that uses the features $\text{ENC}_i(s)$ to predict POS or morphological tags.

### 4.1 Effect of word representation

In this section we compare different word representations extracted with different encoders. Our word-based model uses a word embedding matrix which is initialized randomly and learned with other NMT parameters. For a character-based model we adopt a convolutional neural network (CNN) over character embeddings that is also learned during training (Kim et al., 2015; Costa-jussà and Fonollosa, 2016); see appendix A.1 for specific settings. In both cases we run the encoder over these representations and use its output $\text{ENC}_i(s)$ as features for the classifier.

Table 2 shows POS tagging accuracy using features from different NMT encoders. Char-based models always generate better representations for POS tagging, especially in the case of morphologically-richer languages like Arabic and Czech. We observed a similar pattern in the full morphological tagging task. For example, we obtain morphological tagging accuracy of 65.2/79.66 and 67.66/81.66 using word/char-based representations from the Arabic-Hebrew and Arabic-English encoders, respectively.[2] The superior morphological power of the char-based model also manifests in better translation quality (measured by BLEU), as shown in Table 2.

**Impact of word frequency** Let us look more closely at an example case: Arabic POS and morphological tagging. Figure 3 shows the effect of using word-based vs. char-based feature representations, obtained from the encoder of the Arabic-

---

[2] The results are not far below dedicated taggers (e.g. 95.1/84.1 on Arabic POS/morphology (Pasha et al., 2014)), indicating that NMT models learn quite good representations.

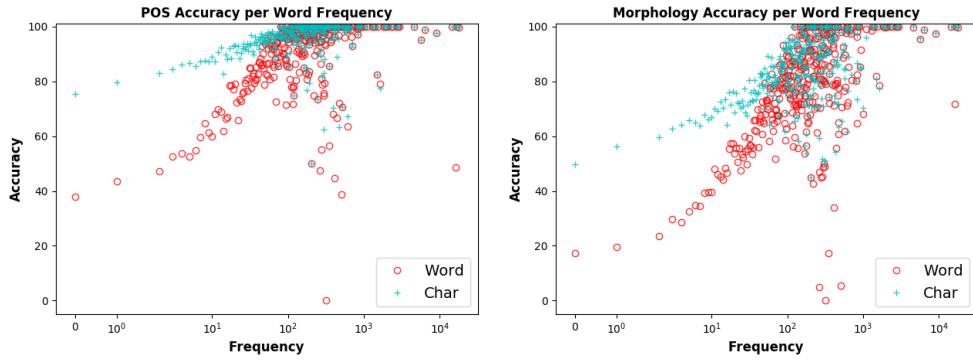

Figure 2: POS and morphological tagging accuracy of word-based and character-based models per word frequency in the training data. Best viewed in color.

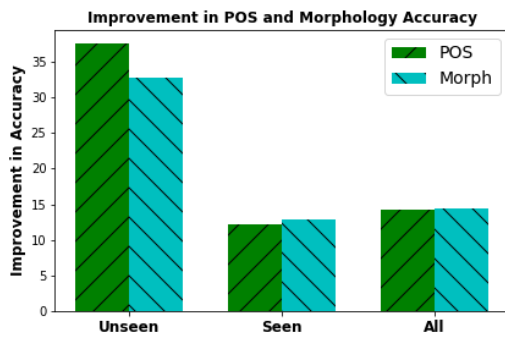

Figure 3: Improvement in POS/morphology accuracy of character-based vs. word-based models for words unseen/seen in training, and for all words.

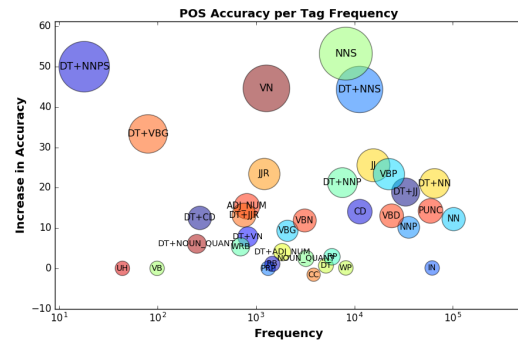

Figure 4: Increase in POS accuracy with char- vs. word-based representations per tag frequency in the training set; larger bubbles reflect greater gaps.

Hebrew system (other language pairs exhibit similar trends). Clearly, the char-based model is superior to the word-based one. This is true for the overall accuracy (+14.3% in POS, +14.5% in morphology), but more so on OOV words (+37.6% in POS, +32.7% in morphology). Figure 2 shows that the gap between word-based and char-based representations increases as the frequency of the word in the training data decreases. In other words, the more frequent the word, the less need there is for character information. These findings make intuitive sense: the char-based model is able to learn character n-gram patterns that are important for identifying word structure, but as the word becomes more frequent the word-based model has seen enough examples to make a decision.

**Analyzing specific tags** In Figure 5 we plot confusion matrices for POS tagging using word-based and char-based representations (from Arabic encoders). While the char-based representations are overall better, the two models still share similar misclassified tags. Much of the confusion comes from wrongly predicting nouns (NN, NNP). In the word-based case, relatively many tags with determiner (DT+NNP, DT+NNPS, DT+NNS, DT+VBG) are wrongly predicted as non-determined nouns (NN, NNP). In the char-based case, this hardly happens. This suggests that char-based representations are predictive of the presence of a determiner, which in Arabic is expressed as the prefix "Al-" (the definite article), a pattern easily captured by a char-based model.

In Figure 4 we plot the difference in POS accuracy when moving from word-based to char-based representations, per POS tag frequency in the training data. Tags closer to the upper-right corner occur more frequently in the training set and are better predicted by char-based compared to word-based representations. There are a few fairly frequent tags (in the middle-bottom part of the figure) whose accuracy does not improve much when moving from word- to char-based representations: mostly conjunctions, determiners, and certain par-

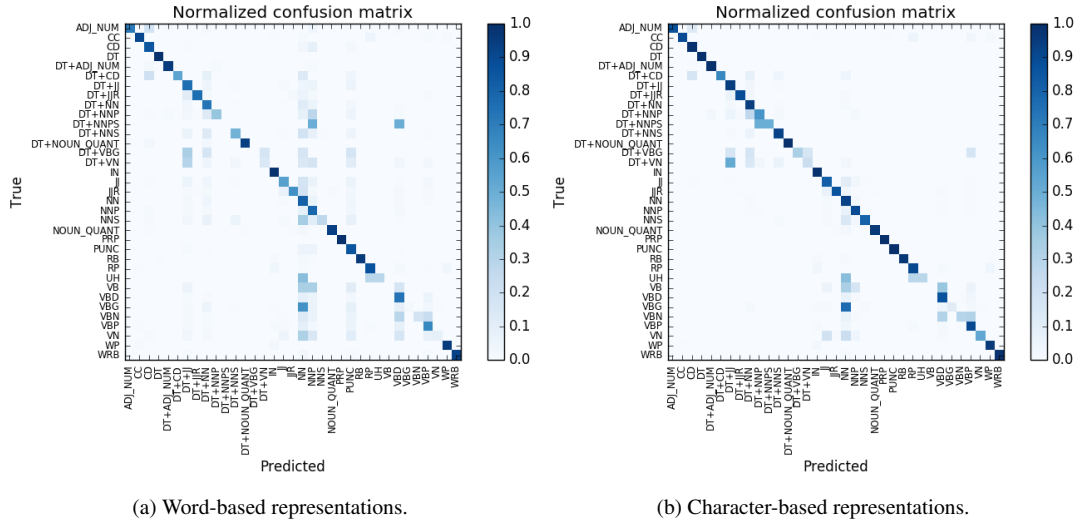

(a) Word-based representations.  (b) Character-based representations.

Figure 5: Confusion matrices for POS tagging using word-based and character-based representations.

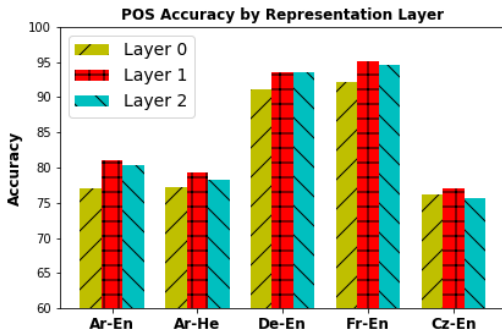

Figure 6: POS tagging accuracy using representations from layers 0 (word vectors), 1, and 2, taken from encoders of different language pairs.

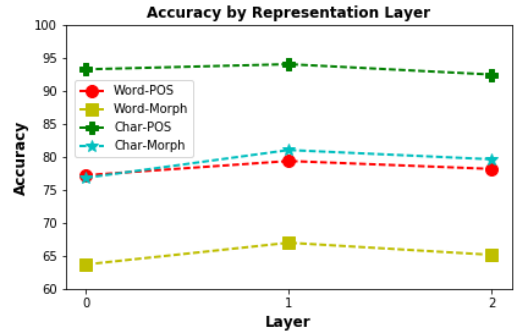

Figure 7: POS and morphological tagging accuracy across layers. Layer 0: word vectors or char-based representations before the encoder; layers 1 and 2: representations after the 1st and 2nd layers.

ticles (CC, DT, WP). But there are several very frequent tags (NN, DT+NN, DT+JJ, VBP, and even PUNC) whose accuracy improves quite a lot. Then there are plural nouns (NNS, DT+NNS) where the char-based model really shines, which makes sense linguistically as plurality in Arabic is usually expressed by certain suffixes ("-wn/yn" for masc. plural, "-At" for fem. plural). The char-based model is thus especially good with frequent tags and infrequent words, which is understandable given that infrequent words typically belong to frequent open categories like nouns and verbs.

## 4.2 Effect of encoder depth

Modern NMT systems use very deep architectures with up to 8 or 16 layers (Wu et al., 2016; Zhou et al., 2016). We would like to understand what kind of information different layers capture. Given

a trained model with multiple layers, we extract representations from the different layers in the encoder. Let $\text{ENC}_i^l(s)$ denote the encoded representation of word $w_i$ after the $l$-th layer. We vary $l$ and train different classifiers to predict POS or morphological tags. Here we focus on the case of a 2-layer encoder-decoder for simplicity ($l \in \{1, 2\}$).

Figure 6 shows POS tagging results using representations from different encoding layers across five language pairs. The general trend is that passing word vectors through the encoder improves POS tagging, which can be explained by contextual information contained in the representations after one layer. However, it turns out that representations from the 1st layer are better than those from the 2nd layer, at least for the purpose of capturing word structure. Figure 7 shows that the same

pattern holds for both word-based and char-based representations, on Arabic POS and morphological tagging. In all cases, layer 1 representations are better than layer 2 representations.[3] In contrast, BLEU scores actually increase when training 2-layer vs. 1-layer models (+1.11/+0.56 BLEU for Arabic-Hebrew word/char-based models). Thus translation quality improves when adding layers but morphology quality degrades. Intuitively, it seems that lower layers of the network learn to represent word structure while higher layers focus more on word meaning. A similar pattern was recently observed in a joint language-vision deep recurrent net (Gelderloos and Chrupała, 2016).

### 4.3 Effect of target language

While translating from morphologically-rich languages is challenging, translating into such languages is even harder. For instance, our basic system obtains BLEU of 24.69/23.2 on Arabic/Czech to English, but only 13.37/13.9 on English to Arabic/Czech. How does the target language affect the learned source language representations? Does translating into a morphologically-rich language require more knowledge about source language morphology? In order to investigate these questions, we fix the source language and train NMT models on different target languages. For example, given an Arabic source we train Arabic-to-English/Hebrew/German systems. These target languages represent a morphologically-poor language (English), a morphologically-rich language with similar morphology to the source language (Hebrew), and a morphologically-rich language with different morphology (German). To make a fair comparison, we train the models on the intersection of the training data based on the source language. In this way the experimental setup is completely identical: the models are trained on the same Arabic sentences with different translations.

Figure 8 shows POS and morphology accuracy of word-based representations from the NMT encoders, as well as corresponding BLEU scores. As expected, translating to English is easier than translating to the morphologically-richer Hebrew and German, resulting in higher BLEU. Despite their similar morphologies, translating Arabic to Hebrew is worse than Arabic to German, which can be attributed to the richer Hebrew morphology

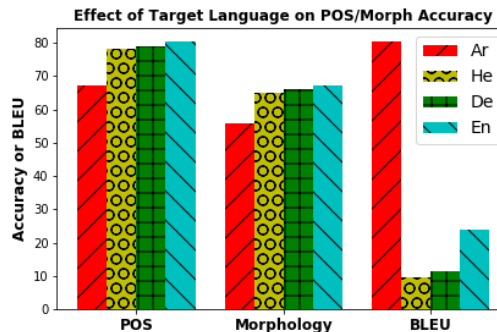

Figure 8: Effect of target language on representation quality of the Arabic source.

compared to German. POS and morphology accuracies share an intriguing pattern: the representations that are learned when translating to English are better for predicting POS or morphology than those learned when translating to German, which are in turn better than those learned when translating to Hebrew. This is remarkable given that English is a morphologically-poor language that does not display many of the morphological properties that are found in the Arabic source. In contrast, German and Hebrew have richer morphologies, so one could expect that translating into them would make the model learn more about morphology.

A possible explanation for this phenomenon is that the Arabic-English model is simply better than the Arabic-Hebrew and Arabic-German models, as hinted by the BLEU scores in Table 2. The inherent difficulty in translating Arabic to Hebrew/German may affect the ability to learn good representations of word structure. To probe this more, we trained an Arabic-Arabic autoencoder on the same training data. We found that it learns to recreate the test sentences extremely well, with very high BLEU scores (Figure 8). However, its word representations are actually inferior for the purpose of POS/morphological tagging. This implies that higher BLEU does not necessarily entail better morphological representations. In other words, a better translation model learns more informative representations, but only when it is actually learning to translate rather than merely memorizing the data as in the autoencoder case. We found this to be consistently true also for char-based experiments, and in other language pairs.

---

[3]We found this result to be also true in French, German, and Czech experiments (see the supplementary material).

| Attn | POS Accuracy | | BLEU | |
|---|---|---|---|---|
| | ENC | DEC | Ar-En | En-Ar |
| ✓ | 89.62 | 43.93 | 24.69 | 13.37 |
| ✗ | 74.10 | 50.38 | 11.88 | 5.04 |

Table 3: POS tagging accuracy using encoder and decoder representations with/without attention.

| | POS Accuracy | | BLEU | |
|---|---|---|---|---|
| | ENC | DEC | Ar-En | En-Ar |
| Word | 89.62 | 43.93 | 24.69 | 13.37 |
| Char | 95.35 | 44.54 | 28.42 | 13.00 |

Table 4: POS tagging accuracy using word-based and char-based encoder/decoder representations.

## 5 Decoder Analysis

So far we only looked at the encoder. However, the decoder DEC is a crucial part in an MT system with access to both source and target sentences. In order to examine what the decoder learns about morphology, we first train an NMT system on the parallel corpus. Then, we use the trained model to encode a source sentence and extract features for words in the target sentence. These features are used to train a classifier on POS or morphological tagging on the target side.[4] Note that in this case the decoder is given the correct target words one-by-one, similar to the usual NMT training regime.

Table 3 (1st row) shows the results of using representations extracted with ENC and DEC from the Arabic-English and English-Arabic models, respectively. There is clearly a huge drop in representation quality with the decoder.[5] At first, this drop seems correlated with lower BLEU in English to Arabic vs. Arabic to English. However, we observed similar low POS tagging accuracy using decoder representations from high-quality models. For instance, the French-to-English model obtains 37.8 BLEU, but its decoder representations give a mere 54.26 accuracy on English POS tagging.

As an alternative explanation for the poor quality of the decoder representations, consider the fundamental tasks of the two NMT modules: encoder and decoder. The encoder's task is to create a generic, close to language-independent representation of the source sentence, as shown by recent evidence from multilingual NMT (Johnson et al., 2016). The decoder's task is to use this representation to generate the target sentence in a specific language. Presumably, it is sufficient for the decoder to learn a strong language model to produce morphologically-correct output, without learning much about morphology, while the encoder needs to learn quite a lot about source language morphol-

---

[4]In this section we only experiment with predicted tags for lack of available parallel data with gold POS/morph. tags.

[5]Decoder results are above a majority baseline of 20%, so the decoder still learns something about the target language.

ogy in order to create a good generic representation. In the following section we show that the attention mechanism also plays an important role in the division of labor between encoder and decoder.

### 5.1 Effect of attention

Consider the role of the attention mechanism in learning useful representations: during decoding, the attention weights are combined with the decoder's hidden states to generate the current translation. These two sources of information need to jointly point to the most relevant source word(s) and predict the next most likely word. Thus, the decoder puts significant emphasis on mapping back to the source sentence, which may come at the expense of obtaining a meaningful representation of the current word. We hypothesize that the attention mechanism hurts the quality of the target word representations learned by the decoder.

To test this hypothesis, we train NMT models with and without attention and compare the quality of their learned representations. As Table 3 shows (compare 1st and 2nd rows), removing the attention mechanism decreases the quality of the encoder representations, but improves the quality of the decoder representations. Without attention, the decoder is forced to learn more informative representations of the target language.

### 5.2 Effect of word representation

We also conducted experiments to verify our findings regarding word-based versus character-based representations on the decoder side. By character representation we mean a character CNN on the input words. The decoder predictions are still done at the word-level, which enables us to use its hidden states as word representations.

Table 4 shows POS accuracy of word- vs. char-based representations in the encoder and decoder. While char-based representations improve the encoder, they do not help the decoder. BLEU scores behave similarly: the char-based model leads to better translations in Arabic-to-English, but not

in English-to-Arabic. A possible explanation for this is that the decoder's predictions are still done at word level even with the char-based model (which encodes the target input but not the output). In practice, this can lead to generating unknown words. Indeed, in Arabic-to-English the char-based model reduces the number of generated unknowns in the test set by 25%, while in English-to-Arabic the number of unknowns remains roughly the same between word- and char-based models.

## 6 Related Work

**Analysis of neural models** The opacity of neural networks has motivated researchers to analyze such models in different ways. One line of work visualizes hidden unit activations in recurrent neural networks that are trained for a given task (Elman, 1991; Karpathy et al., 2015; Kádár et al., 2016; Qian et al., 2016a). While such visualizations illuminate the inner workings of the network, they are often qualitative in nature and somewhat anecdotal. A different approach tries to provide quantitative analysis by correlating parts of the neural network with linguistic properties, for example by training a classifier to predict features of interest. Different units have been used, from word embeddings (Köhn, 2015; Qian et al., 2016b), through LSTM gates or states (Qian et al., 2016a), to sentence embeddings (Adi et al., 2016). Our work is most similar to Shi et al. (2016), who use hidden vectors from a neural MT encoder to predict syntactic properties on the English source side. In contrast, we focus on representations in morphologically-rich languages and evaluate both source and target sides across several criteria. Vylomova et al. (2016) also analyze different representations for morphologically-rich languages in MT, but do not directly measure the quality of the learned representations.

**Word representations in MT** Machine translation systems that deal with morphologically-rich languages resort to various techniques for representing morphological knowledge, such as word segmentation (Nießen and Ney, 2000; Koehn and Knight, 2003) and factored translation models (Koehn and Hoang, 2007). Characters and other sub-word units have become increasingly popular in neural MT, although they had also been used in phrase-based MT (Luong et al., 2010). Such units can be obtained in a pre-processing step – e.g. by byte-pair encoding (Sennrich et al., 2016) or the word-piece model (Wu et al., 2016) – or learned during training with a character-based convolutional/recurrent sub-network (Costa-jussà and Fonollosa, 2016; Luong and Manning, 2016; Vylomova et al., 2016). The latter approach has the advantage of keeping the original word boundaries without requiring pre- and post-processing. Here we focus on a character CNN which has been used in language modeling and machine translation (Kim et al., 2015; Costa-jussà and Fonollosa, 2016; Jozefowicz et al., 2016). We evaluate the quality of different representations learned by an MT system augmented with a character CNN in terms of POS and morphological tagging, and contrast them with a purely word-based system.

## 7 Conclusion

Neural nets have become ubiquitous in machine translation due to their elegant architecture and good performance. The representations they use for linguistic units are crucial for obtaining high-quality translation. In this work, we investigated how neural MT models learn word structure. We evaluated their representation quality on POS and morphological tagging in a number of languages. Our results lead to the following conclusions:

- Character-based representations are better than word-based ones for learning morphology, especially in rare and unseen words.

- Lower layers of the neural network are more focused on word structure, while higher ones are better for learning word meaning.

- Translating into morphologically-poorer languages leads to better source-side representations. This is partly correlated with BLEU.

- The attentional decoder learns impoverished representations that do not carry much information about morphology.

These insights can guide further development of neural MT systems. For instance, jointly learning translation and morphology can possibly lead to better representations and improved translation. Our analysis indicates that this kind of approach should take into account factors such as the encoding layer and the type of word representation.

Another area for future work is to extend the analysis to other representations (e.g. byte-pair encoding), deeper networks, and more semantically-oriented tasks such as semantic parsing.

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
