# Peer review of "What do Neural Machine Translation Models Learn about Morphology?"

_ACL 2017 — decision unknown_

[Official Review · Reviewer 1 · rating 4 · confidence 4]
soundness 5 · originality 5 · clarity 5 · impact 3 · substance 3 · appropriateness 5 · meaningful comparison 3 · presentation format Poster

- Strengths: The authors have nice coverage of a different range of language
settings to isolate the way that relatedness and amount of morphology interact
(i.e., translating between closely related morphologically rich languages vs
distant ones) in affecting what the system learns about morphology. They
include an illuminating analysis of what parts of the architecture end up being
responsible for learning morphology, particularly in examining how the
attention mechanism leads to more impoverished target side representations.
Their findings are of high interest and practical usefulness for other users of
NMT. 

- Weaknesses: They gloss over the details of their character-based encoder.
There are many different ways to learn character-based representations, and
omitting a discussion of how they do this leaves open questions about the
generality of their findings. Also, their analysis could've been made more
interesting had they chosen languages with richer and more challenging
morphology such as Turkish or Finnish, accompanied by finer-grained morphology
prediction and analysis.

- General Discussion: This paper brings insight into what NMT models learn
about morphology by training NMT systems and using the encoder or decoder
representations, respectively, as input feature representations to a POS- or
morphology-tagging classification task. This paper is a straightforward
extension of "Does String-Based Neural MT Learn Source Syntax?," using the same
methodology but this time applied to morphology. Their findings offer useful
insights into what NMT systems learn.

[Official Review · Reviewer 2 · rating 4 · confidence 4]
soundness 5 · originality 5 · clarity 5 · impact 3 · substance 4 · appropriateness 5 · meaningful comparison 3 · presentation format Oral Presentation

Strengths:

- This paper describes experiments that aim to address a crucial
problem for NMT: understanding what does the model learn about morphology and
syntax, etc..
- Very clear objectives and experiments effectively laid down.              Good
state
of the art review and comparison. In general, this paper is a pleasure to read.
- Sound experimentation framework. Encoder/Decoder Recurrent layer
outputs are used to train POS/morphological classifiers. They show the effect
of certain changes in the framework on the classifier accuracy (e.g. use
characters instead of words).
- Experimentation is carried out on many language pairs.
- Interesting conclusions derived from this work, and not all agree with
intuition.

Weaknesses:

 -  The contrast of character-based vs word-based representations  is slightly
lacking: NMT with byte-pair encoding is showing v. strong performance in the
literature. It would have been more relevant to have BPE in the mix, or replace
word-based representations if three is too many.
 - Section 1: "… while higher layers are more focused on word meaning";
similar sentence in Section 7. I am ready to agree with this intuition, but I
think the experiments in this paper do not support this particular sentence.
Therefore it should not be included, or it should be clearly stressed that this
is a reasonable hypothesis based on indirect evidence (translation performance
improves but morphology on higher layers does not).

Discussion:

This is a  fine paper that presents a thorough and systematic analysis of the
NMT model, and derives several interesting conclusions based on many data
points across several language pairs. I find particularly interesting that (a)
the target language affects the quality of the encoding on the source side; in
particular, when the target side is a morphologically-poor language (English)
the pos tagger accuracy for the encoder improves. (b) increasing the depth of
the encoder does not improve pos accuracy (more experiments needed to determine
what does it improve); (c) the attention layer hurts the quality of the decoder
representations.  I wonder if (a) and (c) are actually related? The attention
hurts the decoder representation, which is more difficult to learn for a
morphologically rich language; in turn, the encoders learn based on the global
objective, and this backpropagates through the decoder. Would this not be a
strong
indication that we need separate objectives to govern the encoder/decoder
modules of
the NMT model?